# Sex Matters–Insights from Testing Drug Efficacy in an Animal Model of Pancreatic Cancer

**DOI:** 10.3390/cancers16101901

**Published:** 2024-05-16

**Authors:** Benjamin Schulz, Emily Leitner, Tim Schreiber, Tobias Lindner, Rico Schwarz, Nadine Aboutara, Yixuan Ma, Hugo Murua Escobar, Rupert Palme, Burkhard Hinz, Brigitte Vollmar, Dietmar Zechner

**Affiliations:** 1Rudolf-Zenker-Institute of Experimental Surgery, Rostock University Medical Center, 18057 Rostock, Germany; benjamin.schulz@med.uni-rostock.de (B.S.); emily.leitner@uni-rostock.de (E.L.); tim.schreiber@med.uni-rostock.de (T.S.); brigitte.vollmar@uni-rostock.de (B.V.); 2Core Facility Multimodal Small Animal Imaging, Rostock University Medical Center, 18057 Rostock, Germany; tobias.lindner@med.uni-rostock.de; 3Institute of Pharmacology and Toxicology, Rostock University Medical Center, 18057 Rostock, Germany; rschwarz-arbeit@gmx.de (R.S.); nadineaboutara@gmail.com (N.A.); burkhard.hinz@med.uni-rostock.de (B.H.); 4Department of Medicine Clinic III, Hematology, Oncology and Palliative Medicine, Rostock University Medical Center, 18057 Rostock, Germany; yixuan.ma@med.uni-rostock.de (Y.M.); hugo.murua.escobar@med.uni-rostock.de (H.M.E.); 5Experimental Endocrinology, Department of Biological Sciences, University of Veterinary Medicine Vienna, 1210 Vienna, Austria; rupert.palme@vetmeduni.ac.at

**Keywords:** PDAC, KRAS, SOS1, PI3K, MEK1/2, sex, small molecule, BI-3406, trametinib, BKM120

## Abstract

**Simple Summary:**

Pancreatic ductal adenocarcinoma continues to be one of the deadliest cancers worldwide. Preclinical studies involving animals rarely include sex as a major biological variable in testing the efficacy of new drugs. In an animal model of pancreatic cancer, we analyzed the impact of sex on the pathological features of the disease and on an experimental small molecule-based therapy tested in vivo for the first time. While the therapy shows potential, the obtained results are confounded by sex-specific effects. This study, therefore, highlights the importance of sex-inclusive research while simultaneously providing a basis for further studies of the therapy tested.

**Abstract:**

Preclinical studies rarely test the efficacy of therapies in both sexes. The field of oncology is no exception in this regard. In a model of syngeneic, orthotopic, metastasized pancreatic ductal adenocarcinoma we evaluated the impact of sex on pathological features of this disease as well as on the efficacy and possible adverse side effects of a novel, small molecule-based therapy inhibiting KRAS:SOS1, MEK1/2 and PI3K signaling in male and female C57BL/6J mice. Male mice had less tumor infiltration of CD8-positive cells, developed bigger tumors, had more lung metastasis and a lower probability of survival compared to female mice. These more severe pathological features in male animals were accompanied by higher distress at the end of the experiment. The evaluated inhibitors BI-3406, trametinib and BKM120 showed synergistic effects in vitro. This combinatorial therapy reduced tumor weight more efficiently in male animals, although the drug concentrations were similar in the tumors of both sexes. These results underline the importance of sex-specific preclinical research and at the same time provide a solid basis for future studies with the tested compounds.

## 1. Introduction

For a long time, male animals and patients were predominantly used in both preclinical and clinical biomedical research [1,2]. Results were often applied to women without adequate justification, leading to potentially serious consequences for women [2,3]. The underrepresentation of women in drug trials became particularly apparent when the U.S. Food and Drug Administration (FDA) withdrew ten prescription drugs from the market between 1997 and 2000, four of which have led to more severe adverse events in women. This included the antihistamine Hismanal, which triggered torsade de pointes [3,4]. To address this issue, the 21st Century Cures Act in 2016 called on the National Institutes of Health (NIH) to revise guidelines for the inclusion of women in clinical trials and outlined requirements for reporting analyses by gender, race, and ethnicity in phase III clinical trials [5]. Today, women represent 50% or more of participants in NIH-funded studies [6].

While progress has been made in incorporating female participants in clinical trials, the extension of this inclusion to animal experiments is still insufficient [6,7]. The omission of females from preclinical research was motivated by concerns among researchers about potential variations induced by the estrus cycle, along with apprehensions about increased costs [8,9]. However, since hormonal differences play an important role not only in the development of diseases such as anxiety and depression, asthma, and inflammatory bowel disease but also in pharmacokinetics, it was important to establish guidelines that define sex as a biological variable [10,11,12,13,14]. Therefore, in 2016, the Association of Science Editors published the SAGER guidelines (Sex and Gender Equity in Research), which regulate reporting on sex and gender in scientific publications [2,15].

The lack of sex-specific investigations in preclinical and clinical studies becomes particularly evident within the field of cancer and cancer therapeutics [16]. An evolving awareness emphasizes that sex not only shapes susceptibility to cancer but also influences disease progression and responses to therapeutic interventions [17,18]. In general, men exhibit elevated cancer incidence and mortality rates compared to women [17,18]. Sex hormones have been identified as influential factors affecting both tumor growth and the tumor microenvironment [19,20]. In addition, females exhibit heightened innate and adaptive immune responses, resulting in a lower cancer incidence compared to males and a different response to immunotherapies such as immune checkpoint inhibitors [19,21,22,23]. While men appear to respond better to immune checkpoint inhibitors, women tend to respond better to multimodal immunotherapies [23,24]. For example, a 2021 study by Kim et al. investigated the impact of sex when treating pancreatic cancer patients with FOLFIRINOX [25]. This study reported a markedly enhanced overall survival among females compared to males despite a more rapid dose reduction during each treatment cycle.

Pancreatic ductal adenocarcinoma (PDAC) represents an enormous challenge due to its heterogeneity, plasticity, and aggressive biological nature, making it the fourth leading cause of cancer-related mortality with a 5-year survival rate of approximately 10% [26]. The most frequently affected oncogene, at around 90%, is the Kirsten-rat sarcoma viral oncogene homolog *KRAS* [27]. The mutation affects the GTPase activity of KRAS by altering the homeostatic balance of GDP and GTP binding towards the active state [27]. This occurs either by reducing GTP hydrolysis or by increasing the GTP loading rate [27,28]. As a result, intracellular signaling pathways, such as phosphoinositide 3-kinase/protein kinase B (PI3K/AKT) or RAF/MEK/ERK, are activated and carcinogenesis is promoted [29].

An example of novel therapeutic agents successfully targeting specific KRAS mutations is sotorasib, the first FDA-approved drug for the treatment of KRAS G12C-mutated non-small cell lung cancer (NSCLC) [30]. Sotorasib’s mechanism of action is to prevent the exchange of the inactive GDP-bound form to its active GTP-loaded state, thereby deactivating downstream signaling pathways [31]. However, G12C mutations occur only in about 1.5% of PDAC patients, with G12D mutations predominating at 39%, followed by G12V mutations [32]. Hence, pan-KRAS therapies offer a treatment option that simultaneously targets multiple KRAS mutations [33,34].

Hofmann et al. developed BI-3406, a son of sevenless homolog 1 (SOS1) inhibitor, which decreases the formation of GTP-loaded RAS and limits the proliferation in a broad spectrum of KRAS-driven cancers [35]. BI-3406 also blocks the feedback activation of RAS signaling observed after the inhibition of MEK by trametinib. A combination of BI-3406 with trametinib consequently led to a strong regression in KRAS-driven tumors [35]. However, tumor growth stasis was observed in colorectal and pancreatic PDX models following the combination of SOS1/MEK inhibitors, suggesting further feedback mechanisms and the need for triple combination therapies to effectively disrupt KRAS signaling [35]. In a previous study, we observed a considerable reduction in the growth of several human pancreatic cancer cell lines when buparlisib (BKM120, a pan-PI3K inhibitor) was combined with BI-3406 and trametinib [36].

This study aimed to assess the efficacy and potential adverse side effects of a combinatorial therapy using BI-3406, trametinib, and BKM120 in a pancreatic ductal adenocarcinoma (PDAC) mouse model. In addition, the inclusion of male and female mice allowed us to evaluate if sex influences cancer progression and therapeutic response.

## 2. Materials and Methods

### 2.1. Animals

C57BL/6J mice used in this study were bred under specified pathogen-free conditions (SPF) in our animal facility. The health status of the animal stock was routinely checked (*Helicobacter* sp., *Rodentibacter heylii*, and murine norovirus were detected in a few mice; these animals were not used). For the duration of the experiment, all mice were single-housed in type III cages (Zoonlab GmbH, Castrop-Rauxel, Germany) with a 12 h light-dark cycle, a temperature of 21 ± 2 °C and relative humidity of 60 ± 20% with food (pellets, 10 mm, ssniff-Spezialdiäten GmbH, Soest, Germany) and tap water ad libitum. Enrichment was provided by nesting material (shredded tissue paper, Verbandmittel GmbH, Frankenberg, Germany), a paper roll (75 × 38 mm, H 0528–151, ssniff-Spezialdiäten GmbH), and a wooden stick (40 × 16 × 10 mm, Abedd, Vienna, Austria). All animal experiments were approved by the German local authority: Landesamt für Landwirtschaft, Lebensmittelsicherheit und Fischerei Mecklenburg-Vorpommern (AZ: 1-016/21).

### 2.2. Cell Culture and In Vitro Analysis

The 6606PDA cell line was a kind gift from Prof. Tuveson (Cold Spring Harbor Laboratory, Cold Spring Harbor, NY, USA) [37]. The *Kras* G12D mutation was verified by sequencing and the sex of the cell line was determined by Jarid1c/d PCR [38] to be male (Appendix A). Sanger sequencing of the *Kras* PCR products was carried out by LGC Genomics GmbH (Berlin, Germany) on an ABI 3730 XL DNA Analyzer (Applied Biosystems, Waltham, MA, USA). The cells were routinely cultured in DMEM (4.5 g/L Glucose, PAN Biotech GmbH, Aidenbach, Germany) supplemented with 10% fetal calf serum (FCS, PAN Biotech GmbH, Aidenbach, Germany) and penicillin/streptomycin (100 U/ml, PAN Biotech GmbH, Aidenbach, Germany). To analyze the effects of BI-3406 (a kind gift from Boehringer Ingelheim, supplied through their Open Innovation Portal opnMe), trametinib and BKM120 (both bought from Chemietek, Indianapolis, IN, USA) on proliferation and cell death alone and in combination, cells were seeded at a density of 2 × 10^3^ (proliferation) in 96-well plates (Greiner Bio-One GmbH, Frickenhausen, Germany) or 3 × 10^4^ (cell death) in 12-well plates (Greiner Bio-One GmbH, Frickenhausen, Germany). The substances were dissolved in DMSO and were added at the indicated concentrations either alone or in combination and the cells were incubated for 48 h. Cell proliferation was assessed by the addition of 5-bromo-2′-deoxyuridine (BrdU, Merck KGaA, Darmstadt, Germany) and quantified with a colorimetric cell proliferation ELISA kit (Roche Diagnostics, Mannheim, Germany) according to the manufacturer’s recommendations. The absorbance was measured on a Perkin Elmer Victor X3 model 2030 Multilabel Plate Reader (PerkinElmer, Waltham, MA, USA). Cell death (apoptosis and necrosis) was assessed by the addition of Annexin-V-FITC (BD Biosciences, Heidelberg, Germany) and propidium iodide (Merck KGaA, Darmstadt, Germany) after incubation with test substances. Subsequently, the cells were analyzed by flow cytometry (FACSVerse^TM^, BD Biosciences, Heidelberg, Germany). Inhibitor interaction for proliferation and cell death was evaluated by use of the Bliss independent model as described before [36]. The interaction of the inhibitors was determined by the difference between the observed (E_O_) and predicted (E_P_) effect of the combination and deemed synergistic when E_O_ > E_P_, additive when E_O_ = E_P_ and antagonistic when E_O_ < E_P_. For the analysis of the effects of the physiological and supraphysiological concentrations of sex hormones (testosterone (86500), dihydrotestosterone (10300), progesterone (P8783) and 17β-estradiol (E8875), all bought from Merck KGaA, Darmstadt, Germany), 6606PDA cells were seeded at a density of 5 × 10^3^ and cultured as described before with either 2% or 10% FCS and the sex hormones added at the indicated concentrations for 48 h. Cytotoxicity and cell viability were assessed by CellTox™ Green cytotoxicity assay and CellTiter-Glo^®^ luminescent cell viability assay (both from Promega GmbH, Walldorf, Germany). To evaluate the impact of sex hormones on the efficacy of the combinatorial therapy at physiological concentrations, 6606PDA cells were cultured as described before and BI-3406, trametinib, BKM120, and the respective sex hormones were added at the indicated concentrations and incubated for 48 h. Cytotoxicity was assessed as described before.

### 2.3. Pancreatic Cancer Model and Therapeutic Intervention

Female and male C57BL/6J mice, between 16 and 23 weeks old, were anesthetized with 1–3 vol.% isoflurane. Carprofen (5 mg/kg) was injected subcutaneously as perioperative analgesia and eye ointment was applied. During surgery, the mice were kept warm on a heating plate at 37 °C. Each animal received an orthotopic and an intravenous cell injection. For intravenous injections, 6606PDA cells were resuspended in Hank’s Balanced Salt Solution (HBSS, PAN Biotech GmbH, Aidenbach, Germany); 50 µL of the cell suspension (7 × 10^6^ cells/mL) were injected into the tail vein through a catheter (Fine Bore Polyethylene Tubing (0.28 mm ID, 0.61 mm OD), Smiths Medical International Ltd., Hythe, UK). The orthotopic injection of tumor cells was performed as previously described [39,40,41]. Briefly, the abdomen was shaved, opened and the 6606PDA cells (2.5 × 10^5^ cells in 5 μL PBS/Matrigel (BD Basement Membrane Matrix (354248), Corning Inc., New York, NY, USA) were injected with a 25 μL syringe (Hamilton, Reno, NV, USA) into the head of the pancreas. Afterward, the abdomen was closed with two sutures (Johnson & Johnson Medical GmbH, Norderstedt, Germany) and mice were placed in front of a heating lamp for 20–30 min; 3000 mg/L Metamizol (Novaminsulfon-ratiopharm 500 mg/mL, Ratiopharm GmbH, Ulm, Germany) was added daily to the drinking water for continuous analgesia until the end of the experiment. On day 4 after surgery, the animals were randomized into treatment groups (either vehicle or a combination of BI-3406, trametinib, and BKM120). The compounds were dissolved in a mixture of 60% Phosal50PG (Lipoid GmbH, Ludwigshafen, Germany), 30% PEG400 (Merck KGaA, Darmstadt, Germany) and 10% Ethanol (99.6%, undenatured). Test substances and vehicle were administered by oral gavage from day 4 until euthanasia at day 36 in a 5-days on/2-days off dosing scheme. The drug concentrations administered per gavage were as follows: BI-3406 at 50 mg/kg (2 × per day), trametinib at 0.1 mg/kg (2 × per day), and BKM120 at 30 mg/kg (1 × per day). Sham treatment was performed with the vehicle solution. On day 36 after tumor cell injection mice were gavaged (vehicle or drugs) and injected with BrdU (2.5 µL/g body weight at a concentration of 20 mg/mL) 1.5–2 h before euthanasia by cervical dislocation in deep narcosis (4–5 vol.% isoflurane). Tumors, lungs, livers, and kidneys were harvested and preserved in either 4% PBS-buffered paraformaldehyde (PFA, Formafix GmbH, Düsseldorf, Germany), TissueTek^TM^ (Sakura Finetek Germany GmbH, Umkirch, Germany) or snap frozen in liquid nitrogen for later analysis. Sixty-four mice were used in total for all experiments, of which eight mice had to be excluded due to perioperative complications. 

### 2.4. Assessment of Animal Wellbeing

In order to evaluate animal wellbeing, the body weight, burrowing activity, nesting behavior, distress score, and fecal corticosterone metabolites (FCMs) were assessed for each mouse at distinct time points. For example, the distress score was evaluated on day 0 (30 min after finishing surgery), burrowing and nesting activity was assessed from the evening of day 0 to the morning of day 1, and on day 1 after surgery the body weight was determined and the feces were collected. In order to obtain an overview of animal well-being over the course of the experiment, all the parameters were assessed before the cell injection (day −4 to −3), during the acute (day 0 to 1), early (day 4 to 5), middle (day 18 to 19) and late phase (day 35 to 36) of the experiment. The burrowing activity was analyzed using a 3D-printed tube (length: 15 cm, diameter: 6.5 cm) filled with 200 g of food pellets. The tube was placed into the mouse cage 2–3 h before the dark phase and the remaining pellets were weighed after 17 h on the next day. To analyze the nest-building behavior, a cotton nestlet (5 cm square of pressed cotton batting, Zoonlab GmbH, Castrop-Rauxel, Germany) was placed into the cage 30 to 60 min before the dark phase. The nests were scored at 9:30 a.m. ± 2 h on the next day by using a scoring system developed by Deacon [42]. A sixth score point was added to this scoring system, which defines a perfect nest: The nest looked like a crater and more than 90% of the circumference of the nest wall was higher than the body height of the coiled-up mouse. In addition, the wellbeing of mice was evaluated by assessing multiple parameters with the help of a score sheet previously published [43]. The score summarizes various defined criteria (e.g., spontaneous behavior, flight behavior, or general body conditions). In order to assess the concentration of fecal corticosterone metabolites [44], feces dropped within 24 h in a new cage were collected in every phase of the experiment and stored at −80 °C. Before extraction, the fecal pellets were dried for 4 h at 65 °C and stored at −20 °C. Afterward, 50 mg of the dry feces were extracted with 1 mL 80% methanol for subsequent analysis using a 5α-pregnane-3β,11β,21-triol-20-one enzyme immunoassay [44,45,46].

### 2.5. Tumor Volume (MRI)

For the quantification of tumor volume in vivo, a subset of both female and male mice was scanned 2–7 days before euthanasia with a 7 T MRI (magnetic resonance imaging, BioSpec 70/30, 7.0 Tesla, gradient insert: BGA-12S HP, transmit volume resonator (86 mm inner diameter) and receive-only 2x2-array surface coil (all Bruker BioSpin GmbH, Ettlingen, Germany). For scanning, animals were anesthetized with 1.0–2.5 vol.% isoflurane and were placed in a supine position on the bed of the scanner. The scanning protocol comprised three orthogonal morphological T2-weighted TurboRARE (Rapid Acquisition with Relaxation Enhancement) sequences with the parameters specified in Appendix A. Tumor volume was further quantified in the axial slice faction with the software program ITK-SNAP 3.8.0 [47]. During all imaging procedures, the breathing rate and body temperature were monitored and the temperature of the animals was kept constant by a heating pad. All sequences were triggered by respiration.

### 2.6. Concentration of Tested Compounds In Vivo (LC-MS/MS)

Prior to an analysis by LC-MS/MS, all samples were subjected to appropriate work-up in order to be able to analyze trametinib, BI-3406 and BKM120 in one single run. Therefore, 10 µL mouse plasma or shredded mouse tissue (tumor, liver, kidney) was mixed with 80 µL acetonitrile; 10 µL of a 0.5 µM acridine orange solution was added as an internal standard. After vortexing the samples for 5 min, they were centrifuged at 14,000 rpm (26,342 g) for 5 min; 2 µL supernatant was injected for LC-MS/MS analysis. In order to generate a standard curve for the determination of the concentrations of trametinib, BKM120 and BI-3406 in the mouse samples, the stock solutions of the substances in DMSO were serially diluted using human plasma; 10 µL of these plasma concentrations were processed like the mouse samples. A concentration range of 0.5–5 µM was chosen for BKM120, 0.03125–2.5 µM for trametinib, and 0.25–5 µM for BI-3406. Separation was achieved using a Shimadzu LC-20AD HPLC with a Multospher 120 C18 AQ column 125 × 2 mm, 5 µm particle size (CS-Chromatographie Service GmbH, Langerwehe, Germany) coupled with a guard column (20 mm × 3 mm, 5 µm particle size). Water was chosen as mobile phase A and acetonitrile as mobile phase B, both containing 0.2% formic acid. The flow rate was 0.3 mL/min. A linear gradient from 20% B to 100% B within 4.5 min was chosen for the separation. This step was held for 0.5 min, then immediately reduced to 20% B and the column re-equilibrated for a further 3 min. The total run time was 8 min per run. The oven temperature was set to 40 °C. Mass spectrometric analysis was carried out on a Shimadzu LCMS-8050 triple quadrupole mass spectrometer. The substances were measured in positive or negative mode. The respective transitions and associated mass spectrometric settings as well as the general settings of the triple quadrupole are shown in Appendix A. Calculation of the concentration of the tested compounds was conducted using the volume quantified by MRI-scanning to arrive at the molar concentrations in the tumor.

### 2.7. Histology

After fixation for at least 24 h in 4% PBS-buffered PFA the left lung lobe was serially cut into 4 µm sections and stained with hematoxylin and eosin to assess metastasis. The metastatic area was evaluated with QuPath 0.4.3. Tumors were fixed similarly and immunohistochemistry was subsequently performed for CD8α (1:100, 4SM15-Biotin, eBioscience, San Diego, CA, USA) with secondary detection by Streptavidin-AP (1:100, Invitrogen, Waltham, MA, USA), anti-BrdU (1:50, BU20a, Dako, Hamburg, Germany) with secondary detection by a HRP-conjugated antibody (polyclonal goat anti-mouse, 1:100, Dako, Hamburg, Germany) or anti-PD-L1 (1:200, D5V3B, Cell Signalling Technology, Danvers, MA, USA) with secondary detection by an AP-conjugated antibody (goat anti-rabbit, 1:200, 97048, Abcam, Waltham, MA, USA). Detection of BrdU-positive (BrdU^+^) and CD8-positive (CD8^+^) cells as well as PD-L1-positive tissue was performed with QuPath 0.4.3.

### 2.8. Quantitative Real-Time Polymerase Chain Reaction (TaqMan RT-qPCR)

Parts of the tumors were snap-frozen in liquid nitrogen during tissue harvest. Total RNA was extracted using QIAzol lysis reagent and the RNeasy Mini Kit (both from Qiagen, Hilden, Germany). Synthesis of cDNA was carried out with 100 ng of extracted RNA using the High-Capacity cDNA Reverse Transcription Kit (Applied Biosystems, Waltham, MA, USA). The calibrator consisted of RNA extracted from the lungs of four healthy female wild-type C57BL/6J mice. TaqMan qPCR was performed on a Bio-Rad IQ5 real-time qPCR system (Bio-Rad Laboratories GmbH, Feldkirchen, Germany) with probes (Applied Biosystems, Waltham, MA, USA) for *Gapdh* (Mm99999915_g1), *Ipo8* (Mm01255158_m1), *Ifn-γ* (Mm99999071_m1), *Il10* (Mm00439614_m1), *Il2* (Mm00434256_m1), *Tnf-α* (Mm00443258_m1) and *Il6* (Mm99999064_m1) and the cycling parameters detailed in Appendix A. *Gapdh* and *Ipo8* served as reference genes. Ct values were calculated using the QuantStudio software (Version 2.1, Applied Biosystems, Waltham, MA, USA). The data were calculated as ∆Ct (Ct^Avg·Ref (*Gapdh*+*Ipo8*)^ − Ct^GOI^) and ∆∆Ct (∆Ct^Calibrator^ − ∆Ct). Statistics have been analyzed using the ∆∆Ct values.

### 2.9. Blood Chemistry

Blood was drawn immediately before euthanasia by retroorbital bleeding. The blood was subsequently centrifuged and the resulting plasma was stored at −80 °C for later analysis. Parameters of blood chemistry (AST, ALT, creatinine, LDH) were quantified on a cobas c111 (Roche Diagnostics, Mannheim, Germany) and c-peptide was quantified with a mouse c-peptide ELISA kit (ALPCO, Salem, MA, USA) according to the manufacturer’s recommendations. Five healthy animals of either sex have been used as controls.

### 2.10. Data Presentation and Statistical Analysis

All data were analyzed and graphed with GraphPad Prism (version 8.0.1, GraphPad Software Inc., San Diego, CA, USA) and are presented as box plots (single data points are depicted and whiskers indicate minimum and maximum) or as bar graphs. Statistical significance was determined by different methods (for details see figure legends) based on the number of independent variables and data characteristics. If the influence of two independent variables (e.g., time and therapy) on one dependent variable (when using combined data) was analyzed, a two-way repeated measure ANOVA with Sidak’s post-hoc test was performed. If the influence of one independent variable on a dependent variable was evaluated, the normality of data was checked by the Shapiro–Wilk normality test. When two groups were compared and data were not paired, either the unpaired *t*-test (with Welch’s correction if sample sizes were unequal) or the Mann–Whitney rank sum test was performed. When analyzing more than two groups, either a one-way ANOVA with Dunnett’s post-hoc test or a Kruskal–Wallis test with Dunn’s post-hoc test was performed. Therapy response was calculated as a normalized quotient of the tumor weight of individual drug-treated animals and the mean tumor weight of vehicle-treated animals.

$$normalized\;therapy\;response=100-\left(\frac{individual\;tumor\;weight\left(treated\right)}{mean\;tumor\;weight\left(vehicle\right)}\right)\times 100$$


The correlation between CD8^+^ cells and tumor weight was assessed with Spearman correlation and linear regression including confidence intervals. Survival was evaluated by the Kaplan–Meier estimator followed by a log-rank test. Differences with *p* < 0.05 were considered to be significant. 

## 3. Results

### 3.1. Key Pathological Features of the PDAC Model Are Sex-Dependent

Out of 15 male mice, eight had to be euthanized early due to reaching humane endpoints. In contrast, none of the 15 female mice had to be euthanized early and thus their probability of survival was significantly higher than in male mice (Figure 1A). The weight of the tumor in the pancreas was significantly lower in female mice with a median weight of 266 mg compared to male mice with a 625 mg median tumor weight (Figure 1B). The intravenous injection of 6606PDA cells led to the formation of small tumors in the lungs of 33.3% of male animals, while no lesions were detected in female animals (Figure 1C–E and Appendix A). Duodenal invasion of the primary tumor is observed in a subset of animals (Appendix A). In this model, 73.3% of male animals were affected by invasive tumor growth compared to 6.6% of female animals (Figure 1F).

### 3.2. Male Mice Experience More Distress in the Late Phase of the Experiment

During the experiment, parameters of animal wellbeing, such as body weight, burrowing and nesting activity, a clinical distress score, and fecal corticosterone metabolites (FCMs) were assessed. Body weight, burrowing, and the clinical distress score were significantly different between the two sexes in the late phase of the experiment, where the burden of disease is expected to be highest (Figure 2A–C). Body weight and the burrowing activity of male mice were reduced compared to females and the clinical distress score was increased. There was no significant difference in nesting activity and FCMs between male and female mice over the course of the experiment (Figure 2D,E). 

### 3.3. An Experimental Small-Molecule-Based Therapy Shows Promising Results In Vitro

In order to evaluate the inhibitors of KRAS:SOS1, MEK1/2, and PI3K in 6606PDA cells, a combination of 10 µM BI-3406, 0.064 µM trametinib, and 1 µM BKM120 was tested in vitro. These concentrations correspond to approximately 50% of the IC_50_ value of each compound for inhibiting the proliferation of 6606PDA cells (Appendix A). The combinatorial treatment significantly reduced BrdU incorporation in 6606PDA cells when compared to the DMSO control group, single drugs, or a combination of only two drugs (Figure 3A). For all possible combinations of these drugs, the inhibition of proliferation was higher than what would be expected with an additive inhibitory effect (Figure 3B). This demonstrates a synergistic inhibition of proliferation. The combination of these three drugs also induced cell death in a synergistic manner (Figure 3C,D). Physiological [48] and supraphysiological concentrations of sex hormones (testosterone, progesterone, 17β-estradiol, and dihydrotestosterone) have no apparent concentration-dependent effect on the viability and cell death of 6606PDA cells cultured with either 10% or 2% FCS (Appendix A). To evaluate if these sex hormones impact the cytotoxic efficacy of the combinatorial therapy in vitro, 6606PDA cells were treated with the combination of BI-3406, trametinib and BKM120 plus each sex hormone separately (Appendix A). The combination of BI-3406, trametinib, and BKM120, with 17β-estradiol performed significantly worse than the control without sex hormones and the combination with testosterone. However, all the tested sex hormones had a quantifiable negative impact on the efficacy of the combinatorial therapy. 

### 3.4. Sex Impacts the Response to Therapy In Vivo

After confirming the high efficacy of the drugs in vitro, the triple combination of inhibitors was tested in female and male C57BL/6J mice. Since BI-3406, trametinib, and BKM120 had been safely tested in mice before with concentrations of 50 mg/kg, 0.1 mg/kg, and 30 mg/kg, respectively [35,49], we utilized these concentrations in our study. One out of thirteen female mice in the treatment group had to be euthanized, while all vehicle-treated control animals survived until day 36 (Figure 4A). Of 13 drug-treated male mice, six had to be euthanized, while 8 out of 15 vehicle-treated male animals had to be euthanized (Figure 4B). The tumor weight of female animals was only slightly reduced by the compounds with a median weight of 208 mg compared to 266 mg in vehicle-treated animals (Figure 4C). In contrast, the tumor weight of male animals was significantly reduced to a median weight of 99 mg compared to 625 mg in vehicle-treated male mice (Figure 4D). The normalized therapy response was significantly higher in males compared to females (Figure 4E). While invasive tumor growth was detected in 6.6% of untreated female animals, this percentage increased to 15.3% when treated with drugs (Figure 4F). However, invasive tumor growth in male mice was reduced from 73.3% in untreated animals to 46.1% in drug-treated mice (Figure 4G). None of the female mice receiving vehicle solution were affected by lung metastasis, while unexpectedly 30.7% of drug-treated animals had detectable lesions in their left lung lobe (Figure 4H and Appendix A). In a similar manner, male mice showed an increase in metastasis from 33.3% in vehicle-receiving animals to 50% in drug-treated ones (Figure 4I and Appendix A).

To evaluate the concentration of each compound within the tumor, a subset of tumors was subjected to LC-MS/MS analysis. As shown in Table 1, the calculated median concentrations were not markedly different between female and male animals. A comparison with the concentrations used in vitro (Table 2) indicates that the median concentrations of trametinib and BKM120 were higher within the tumor than the concentration used in cell culture, while the median concentration for BI-3406 within the tumor was below the concentration used in vitro. Similarly, concentrations in the plasma, liver and kidney were evaluated in a subset of animals (Appendix A). As before, no marked differences were observed and trametinib could not be detected in the plasma of either sex. 

### 3.5. Impact of Combinatorial Therapy on Parameters Associated with Adverse Side Effects

When assessing potential adverse side effects of the combinatorial therapy, no significant increases in transaminases (AST, ALT), creatinine and lactate-dehydrogenase (LDH) activity were observed in the blood plasma (Figure 5A–H). However, the combination of these drugs significantly increased the c-peptide concentration in both sexes (Figure 5I,J). In addition, the influence of these compounds on the wellbeing of animals was evaluated (Appendix A). The therapy had no significant impact on body weight (Appendix A) and burrowing activity (Appendix A) of either sex. The distress score of drug-treated female mice was significantly increased during the late phase of drug administration (Appendix A). No significant difference in distress scores was observed in male mice (Appendix A). Neither nesting activity (Appendix A) nor FCMs (Appendix A) of both sexes were impacted by these drugs.

### 3.6. Quantification of Tumor Cell Proliferation, CD8^+^ cells and PD-L1 Expression in Males and Female 

With the aim of determining the possible reasons for the marked differences in tumor weight between vehicle-treated female and male mice, we analyzed tumor cell proliferation and the amount of intratumoral CD8^+^ cells in the primary tumor. The median tumor cell proliferation did not differ significantly between the two sexes, with a median of 10% proliferating cells in females compared to 8.9% in males (Figure 6A–C). However, the amount of CD8^+^ cells per mm^2^ was significantly different between female and male animals. Female mice had a median of 639 CD8^+^ cells/mm^2^ tumor tissue compared to 263 CD8^+^ cells/mm^2^ in male animals (Figure 6D–F). In both sexes, there was an inverse correlation between the amount of CD8^+^ cells and the tumor weight (Appendix A). A comparison of relative gene expression of cytokines implicated in T cell activation (*Ifn-γ* and *Il2*) and inflammation (*Il10*, *Il6* and *Tnf-α*) between the sexes revealed a significantly higher relative *Ifn-γ* gene expression in tumors of female mice (Appendix A), while there was no significant difference in the rest of the analyzed cytokines (Appendix A). To assess if the difference in intratumoral CD8^+^ cells between the sexes was accompanied by a difference in PD-L1 expression in the tumor, the PD-L1-positive area in tumors of both sexes was analyzed. Surprisingly, the PD-L1-positive area in tumors was higher in females compared to males (Appendix A). While differentiating between tumor cells and immune cells proved unfeasible in our non-multiplexed IHC setting, positive staining in females (Appendix A) predominantly coincided with inflammatory lesions containing immune cells, which was rarely observed in males (Appendix A).

### 3.7. The Combinatorial Therapy inhibits CD8^+^ Cell Tumor Infiltration and PD-L1 Expression in a Sex-Specific Manner

Similar to the evaluation of tumor cell proliferation and quantity of intratumoral CD8^+^ cells in vehicle-treated animals, both parameters were analyzed in mice treated with combinatorial therapy. The drugs had no significant effect on tumor cell proliferation in both sexes compared to vehicle-treated animals (Figure 7A–D). However, CD8^+^ cells were significantly reduced in females treated with the drugs compared to vehicle-treated ones, while the median amount of CD8^+^ cells in males was barely affected (Figure 7E–H). Moreover, the negative correlation between the CD8^+^ cell count and the tumor weight observed in vehicle-treated mice (Appendix A) disappeared in female drug-treated mice (Appendix A), while it remained intact in male animals treated with these drugs (Appendix A). Additionally, the expression of PD-L1 was analyzed in the tumors. Similarly to the observed effects on CD8^+^ cells, PD-L1 expression was significantly reduced in the tumors of females treated with BI-3406, trametinib, and BKM120 compared to those who received the vehicle solution (Appendix A). In male mice, the therapy had no significant effect on PD-L1 expression (Appendix A). 

## 4. Discussion

We evaluated the influence of sex on the pathological features of a syngeneic, orthotopic and metastasized model of pancreatic cancer. Sex exerted a strong influence on survival, tumor weight, metastasis and tumor invasiveness. Male mice were more severely impacted (Figure 1) and experienced more distress (Figure 2A–C). In addition, a better therapy response was observed in males when using a combination of drugs consisting of BI-3406, trametinib and BKM120 (Figure 4E).

The larger tumor size observed in male mice could neither be attributed to a greater proliferation rate of tumor cells in male mice (Figure 6A–C) nor to a direct influence of sex hormones on cell viability or cell death of 6606PDA cells in vitro, since we have not observed a major reduction in cell viability or increased cytotoxicity when using more than 25 times the physiological concentrations of sex hormones typically found in C57BL/6J mice (Appendix A) [48]. An immunological response to male-specific antigens could also lead to reduced tumor sizes in female mice since the 6606PDA cell line was isolated from a male mouse (Appendix A). In transplantation studies, it was demonstrated that female animals can develop a sex-specific immunological response to male antigens, which plays a prominent role in graft rejection [50,51,52,53]. However, several studies involving syngeneic tumor models show that male mice are more severely impacted, irrespective of the sex of the cell line used [54,55,56,57,58].

While a response to male-specific antigens might not be likely, a stronger adaptive immune response in female mice independent of sex-specific antigens is possible. Indeed, our data show more intratumoral CD8^+^ cells in female tumors, indicating a distinct immune response in males and females (Figure 6D–F). This hypothesis is supported by several studies analyzing the role of CD8^+^ cells in syngeneic tumor models and the impact of sex hormones on adaptive immunity [54,55,56,57,58,59]. For example, Kwon and colleagues demonstrated that injection of bladder cancer cells leads to smaller tumors in female mice compared to male mice, but that difference was reduced by depletion of CD8^+^ cells [56]. In addition, several studies demonstrated that androgens increased tumor volume and at the same time decreased the number of CD8^+^ cells within the tumor [57] or impaired their functionality [56,57,58]. Similar to more intratumoral CD8^+^ cells, female mice also had a higher PD-L1 expression (*p* = 0.0556) in tumors, when compared to males (Appendix A). Expression of PD-L1 was found on single cells and especially in inflammatory lesions within the tumors of female mice (Appendix A). Interestingly, there is clinical evidence, showing that PD-L1 expression on immune cells correlates with improved outcomes in several different cancers [60,61,62,63]. This evidence is consistent with our observation that female mice have smaller tumors and a better survival rate than male mice. Thus, our data as well as the cited literature support the hypothesis that a reduced immune response in males leads to bigger tumors. This is also consistent with epidemiological studies proving the higher incidence and mortality of cancer in males [64].

In addition to analyzing the influence of sex on the pathological features of the cancer model, we also tested the efficacy of an experimental combination of drugs. BI-3406, trametinib, and BKM120 inhibited the proliferation of murine pancreatic cancer cells very efficiently in vitro in a synergistic manner (Figure 3A,B). The clinical relevance of this combinatorial treatment is also supported by a recent study of our group, demonstrating efficacy in several human PDAC cell lines expressing relevant KRAS mutations [36]. However, neither in vivo efficacy nor adverse side effects of this therapy were addressed in that study. In the present study, the therapy reduced the tumor weight with higher efficacy in male mice, accompanied by a modest decrease in tumor invasiveness and increased probability of survival (Figure 4). These promising results suggest that this drug combination might be worth to be tested in additional studies since effective therapies for PDAC are still urgently needed [65].

Furthermore, we investigated whether the drugs caused adverse side effects. To this end, we assessed the parameters of organ damage and animal wellbeing. Compared to vehicle-treated animals, the combination of these drugs had no significant impact on the parameters of liver damage (Figure 5A–D), kidney function (Figure 5E,F), or tissue damage in general (Figure 5G,H). Distress-related parameters (Appendix A) also support the hypothesis that the therapy shows good tolerability in this model. Not unexpectedly, the triple drug combination increased c-peptide concentrations significantly in the blood plasma of both sexes (Figure 5I,J). This is a common phenomenon in clinical trials with BKM120 and pan-PI3K inhibitors, in general, and indicates a limitation of therapies inhibiting this signaling pathway [66]. However, preclinical evidence suggests that this side effect can be ameliorated by metformin or a ketogenic diet [67,68]. In summary, the tested therapy is efficacious in vivo and observed side effects can probably be managed.

However, as is evident from our in vivo data, drug efficacy in female animals is heavily confounded by sex-specific factors (Figure 4E). Indeed, PDAC cells have been found to express sex hormone receptors [69,70] and it has been shown that β-estradiol can sensitize PDAC cells to chemotherapy [71]. Interestingly, we observed the opposite effect in vitro. Especially female sex hormones inhibited the efficacy of the tested drug combination (Appendix A). Thus, it is possible that female sex hormones contribute to the observed reduction in therapy efficacy in vivo. Sex differences in pharmacokinetics and pharmacodynamics have also been reported previously [72,73,74,75]. The proposed mechanisms range from sex-dependent expression levels of efflux transporters and metabolizing enzymes to a direct impact of sex hormones [72,76,77,78,79]. This has a direct influence on the clearance and systemic availability of the administered drugs. However, the systemic drug level is often a poor substitute for the drug concentration at the target site (e.g. in the tumor), which seems to be particularly true for small molecule inhibitors such as those used here [80]. In an attempt to test the hypothesis that the drug efficacy in females is confounded by sex-specific differences in drug concentrations, we subjected tissue and plasma of a subset of animals to analysis by LC/MS-MS. The data show that there is no meaningful difference in the drug concentrations in tumors, as well as plasma, liver, or kidney between male and female animals (Table 1 and Appendix A). Although these data are limited by small sample sizes, the implications are important, as they prove that the compounds are bioavailable and reach their intended target site in males and females alike. Thus, we conclude that other mechanisms must cause higher drug efficacy in male mice. Interestingly, the pathways inhibited by these drugs are also vital for immune cells such as lymphocytes [81,82,83]. Both BKM120 and trametinib are known inhibitors of immune cell proliferation, activation, and effector function [84,85,86,87,88]. Indeed, our data demonstrate that this drug combination leads to immunosuppression primarily in female mice, demonstrated by the occurrence of metastasis in drug-treated females, which was not detected in vehicle-treated animals (Figure 4H and Appendix A). Furthermore, CD8^+^ cells were significantly reduced in female mice receiving therapy compared to those who received vehicle solution (Figure 7G). This was not observed in male mice (Figure 7H). This reduction in CD8^+^ cells was accompanied by a significant decrease in PD-L1 expression in tumors of female mice (Appendix A). In female mice, PD-L1 expression was primarily found in inflammatory lesions within the tumors (Appendix A) and clinical evidence suggests that PD-L1 expression on immune cells correlates with improved outcomes in several different cancers [60,61,62,63]. Based on these findings, we hypothesize that these drugs cause more immunosuppression in female than in male mice. This might prevent a strong reduction in tumor weight by the combinational therapy, an effect clearly observed in male mice (Appendix A). 

The interpretation of the data presented in this study is subject to some limitations. One limitation is the unbalanced sample sizes due to the significantly lower probability of survival of male animals. However, unequal sample sizes are not inherently problematic, as the underlying problem of unequal sample sizes is often unequal variances. Non-parametric tests like Kruskal–Wallis or Mann–Whitney do not make assumptions about equal variances and unequal variances in *t*-tests can be corrected via Welch’s correction, which has been conducted when comparing groups with different sample sizes (Figure 1B and Figure 6C). Yet, ANOVA procedures strongly rely on the assumption of equal variances [89]. As there is no practical alternative for a two-way repeated measure ANOVA [90], we have opted for its use here to interpret animal wellbeing between the two sexes (Figure 2). Thus, these results have to be considered with care and further investigations should strive for equal sample sizes if statistical inference is to be interpreted robustly. 

Another limitation of this study is the use of an orthotopic syngeneic mouse model as the only model. These models are easier to establish and more cost-efficient [91]. However, therapy responses in allograft implantation models can depend on location [92] and used cell lines [93], which leads to interpretations of results being specific to the features of the cell line [94]. There is a multitude of different models used in preclinical PDAC research, ranging from chemically induced models to genetically engineered or implantation models [95,96]. Patient-derived xenograft models are considered a standard in preclinical oncology because the use of human cell lines in treating human cancers is a clear translational advantage [97]. However, these models come with the important disadvantage of immunoincompetence of the host mouse strain, which is a prerequisite to ensure the engraftment of human cells in a different species. Sex differences based on immunological differences, as have been observed in this study, would have been missed in a similar study using immunocompromised strains. Genetically engineered models combine the advantages of immunocompetence and in situ carcinogenesis but significantly prolong experimental duration. Each of these models has advantages but also limitations. Therefore, it is crucial to recognize that the results of this study can be interpreted only within the context of the syngeneic orthotopic mouse model utilized. Replicating these findings across diverse models is essential to confirm their robustness. A retrospective analysis comparing clinical trial outcomes with preclinical data will be required in the future for assessing the translational significance of sex-specific variations observed in preclinical models. This issue holds significant importance, as disregarding sex-specific differences may lead to serious consequences such as inadequate dosing or adverse drug reactions [73,98,99].

## 5. Conclusions

In summary, the novel experimental therapy tested in this study leads to reduced tumor weight. Parameters associated with adverse side effects as well as animal wellbeing indicate that the drugs are well tolerated, with the exception of an observed increase in c-peptide plasma concentrations, which may be controlled by metformin or a ketogenic diet. As novel therapies to combat PDAC are urgently needed, this drug combination offers a promising basis for further studies. Sex-specific effects in female animals confound the obtained results, emphasizing the importance of sex-specific research. These results warrant further in vivo testing of these drugs in other models.

## Figures and Tables

**Figure 1 cancers-16-01901-f001:**
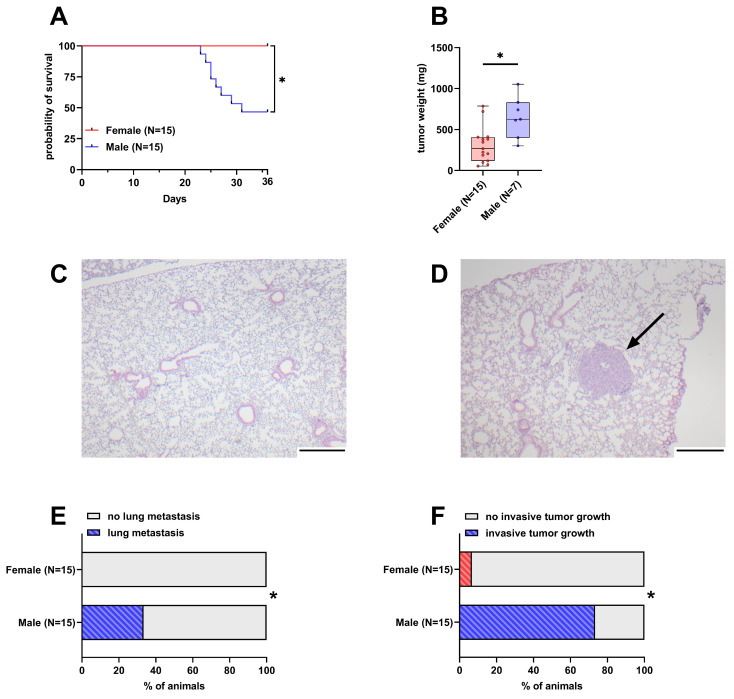
Differences in survival, tumor weight, lung metastasis, and invasive tumor growth between female and male mice. (**A**): Probability of survival in female and male mice (Kaplan–Meier estimator and log-rank Mantel–Cox test, * *p* < 0.05). (**B**): Tumor weight in surviving female and male mice 36 days after orthotopic cell injection (unpaired *t*-test with Welch’s correction, * *p* < 0.05). (**C**,**D**): Representative histological sections of lungs from female (**C**) and male (**D**) mice. The arrow in D highlights a lung metastasis (scale bar = 250 µm). (**E**): Percentage of mice with histologically detectable metastases (Fisher’s exact test, * *p* < 0.05). (**F**): Percent of mice with invasive tumor growth (duodenal invasion, Fisher’s exact test, * *p* < 0.05).

**Figure 2 cancers-16-01901-f002:**
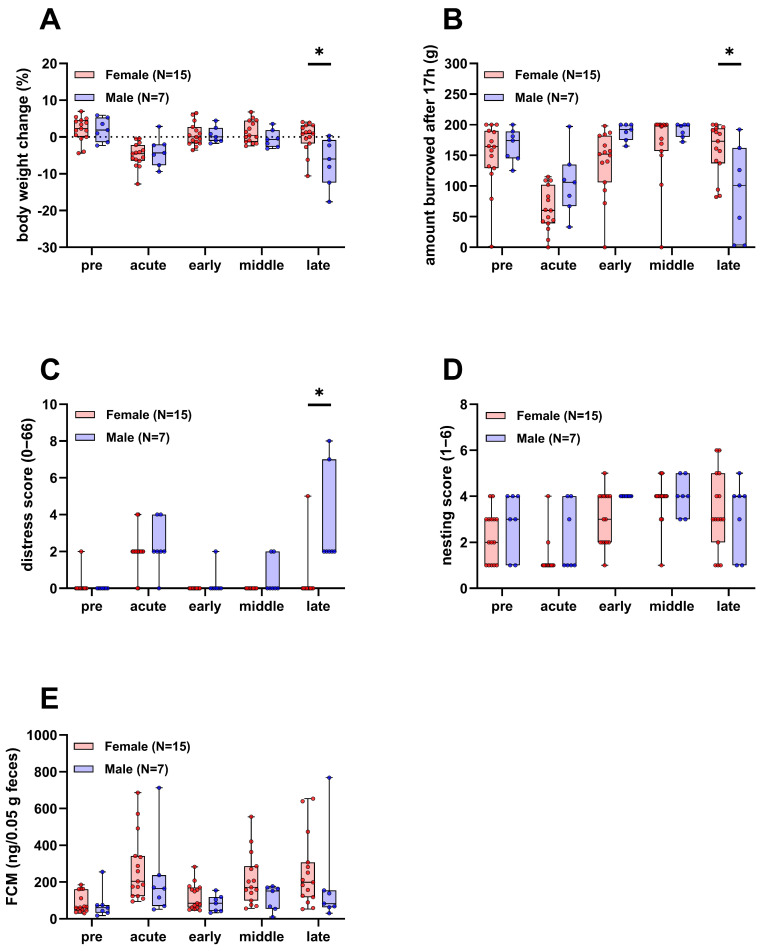
Parameters of animal wellbeing in female and male mice. Body weight (**A**), burrowing activity (**B**), distress score (**C**), nesting activity (**D**), and concentration of fecal corticosterone metabolites (FCMs, (**E**)) compared between surviving female and male mice (Two-way repeated measures ANOVA with Sidak’s post-hoc test, * *p* < 0.05). Pre, acute, early, middle, and late refer to the experimental phases as defined in the methods section.

**Figure 3 cancers-16-01901-f003:**
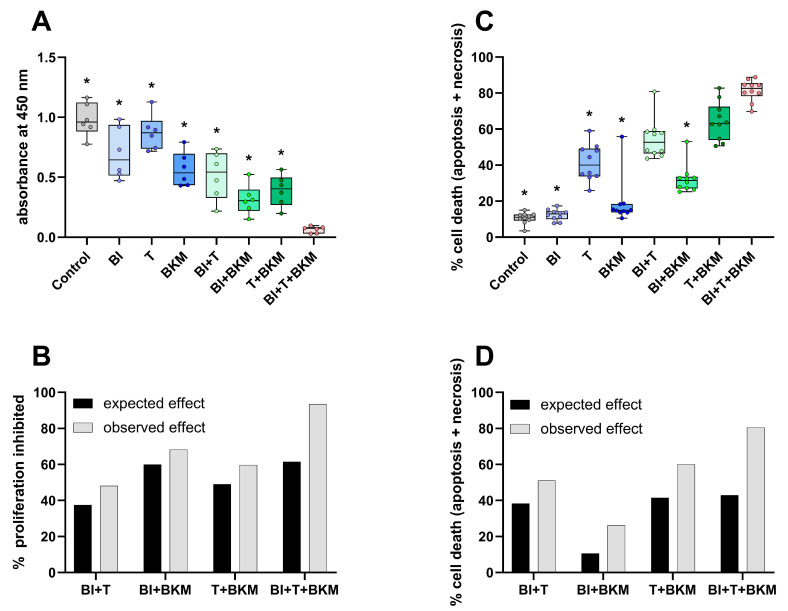
SOS1, MEK1/2 and PI3K inhibitors reduce proliferation and induce cell death. (**A**): Incorporation of BrdU in cells exposed to vehicle control (DMSO), 10 µM BI-3406 (BI), 0.064 µM trametinib (T), 1 µM BKM120 (BKM) or the indicated combinations of these drugs (ordinary one-way ANOVA with Dunnett’s post-hoc test, * *p* < 0.05 compared with triple combination, N = 6). (**B**): The synergy of drugs in inhibiting proliferation was determined using the Bliss independent model. (**C**): Induction of cell death (apoptosis and necrosis, Kruskal–Wallis test with Dunn’s post-hoc test, * *p* < 0.05 compared with triple combination, N = 10). (**D**): Analysis of synergy by the Bliss independent model for induction of cell death for all combinations of the three compounds.

**Figure 4 cancers-16-01901-f004:**
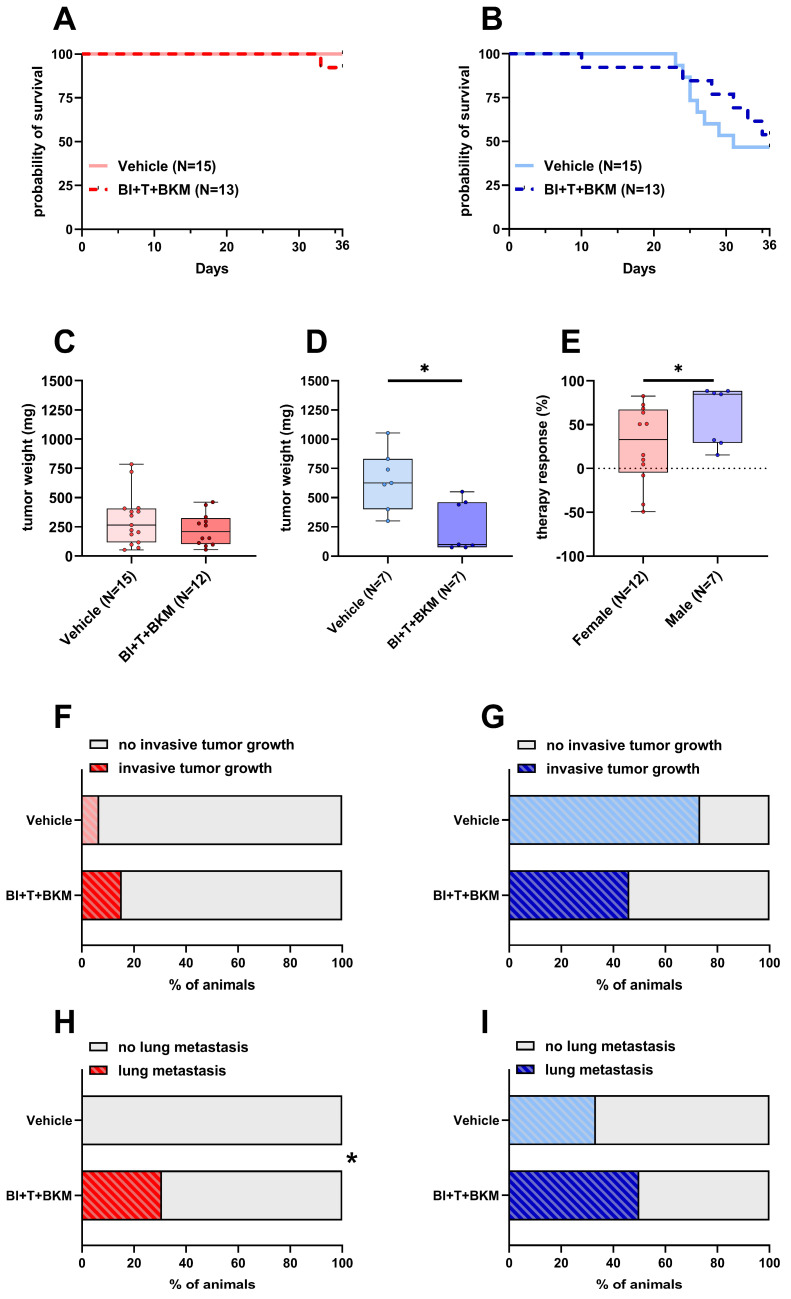
Effects of combinatorial therapy on survival, primary tumor weight, invasive tumor growth and metastasis. (**A**,**B**): Probability of survival in female (**A**) and male (**B**) mice receiving either vehicle or the drug combination consisting of BI-3406 (BI), trametinib (T), and BKM120 (BKM) (Kaplan–Meier estimator and log-rank Mantel–Cox test, ns). (**C**,**D**): Tumor weight of surviving female (**C**) and male (**D**) mice 36 days after tumor cell injection receiving either vehicle or combinatorial therapy ((**C**): unpaired *t*-test, ns; (**D**): Mann–Whitney test, * *p* < 0.05). (**E**): Therapy response of animals receiving therapy normalized to the mean tumor weight of vehicle-treated animals of the same sex (Mann–Whitney test, * *p* < 0.05). (**F**,**G**): Percentage of female (**F**) and male (**G**) animals with invasive tumor growth receiving either vehicle or combinatorial therapy (Fisher’s exact test, ns). (**H**,**I**): Percentage of female (**H**) and male (**I**) animals with detectable lesions in serial histological slices of the left lung lobe receiving either vehicle or combinatorial therapy (Fisher’s exact test, * *p* < 0.05).

**Figure 5 cancers-16-01901-f005:**
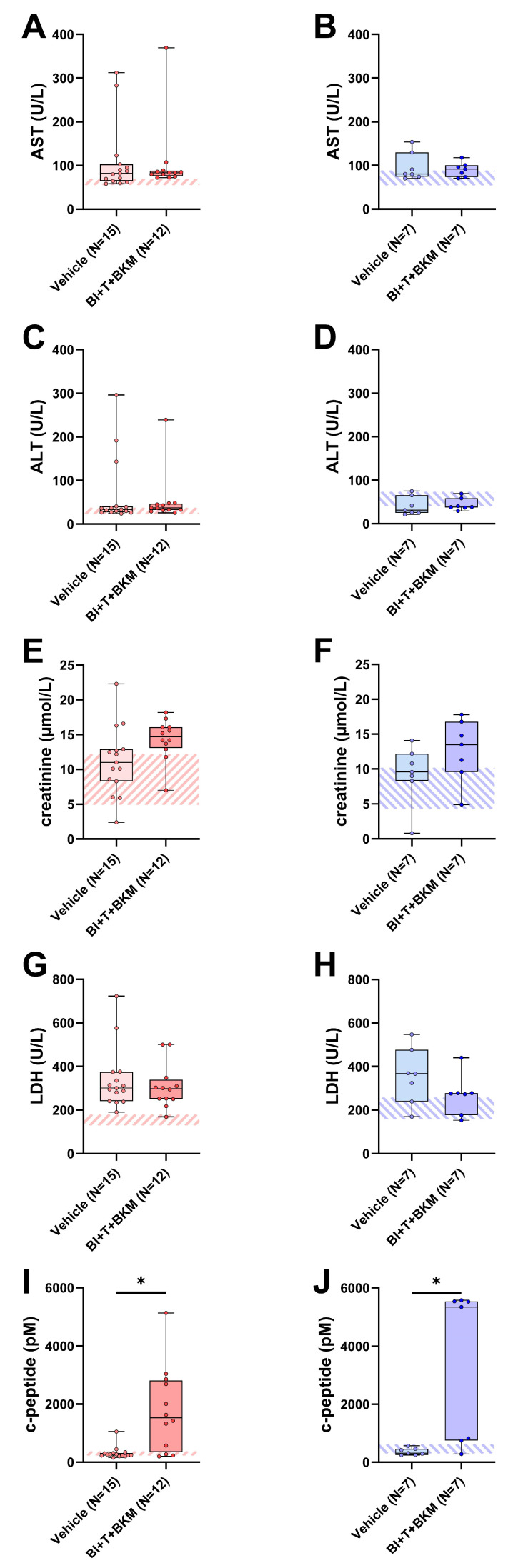
Effect of combinatorial therapy on blood chemistry. Analysis of AST (**A**,**B**), ALT (**C**,**D**), creatinine (**E**,**F**), LDH (**G**,**H**), and c-peptide (**I**,**J**) in blood plasma of surviving female and male mice receiving either vehicle or a combination of drugs consisting of BI-3406 (BI), trametinib (T) and BKM120 (BKM). ((**A**,**C**,**G**,**I**,**J**): Mann–Whitney test, * *p* < 0.05; (**B**,**D**–**F**,**H**): unpaired *t*-test, ns). The striped area indicates the physiological range of each parameter, analyzed from the blood plasma of healthy female or male mice (N = 5 of each sex). For *p*-values of all tested differences see Appendix A.

**Figure 6 cancers-16-01901-f006:**
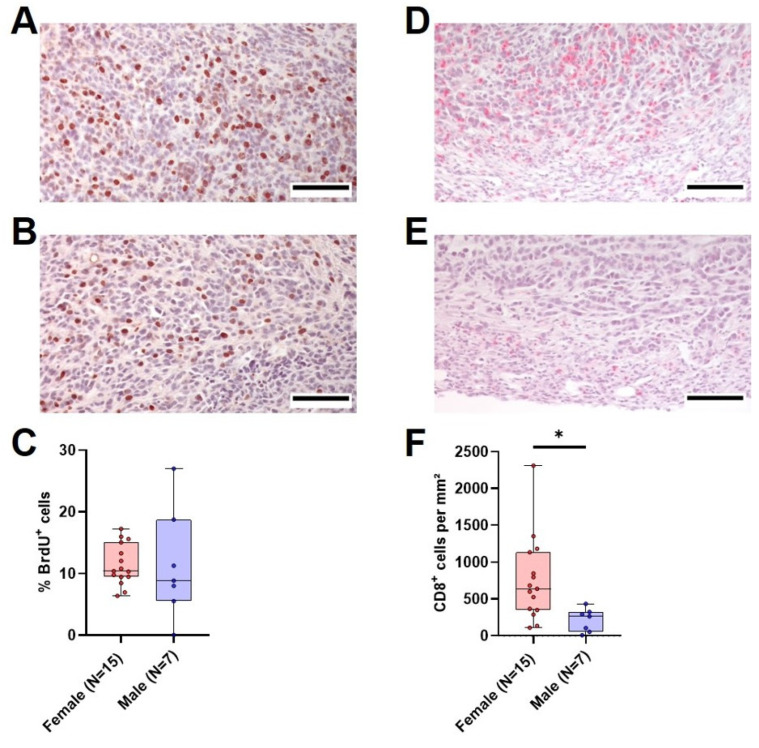
Tumor cell proliferation and intratumoral CD8^+^ cells in vehicle-treated animals. (**A**,**B**): Representative histological sections of anti-BrdU stained (brown nuclei) tumors of surviving female (**A**) and male (**B**) vehicle-treated mice (scale bar = 100 µm) and the quantitative analysis of the percentage of BrdU^+^ cells ((**C**), unpaired *t*-test with Welch’s correction, ns). (**D**,**E**): Representative histological sections of anti-CD8α stained (red) tumors of surviving female (**D**) and male (**E**) vehicle-treated mice (scale bar = 100 µm) and the quantitative analysis of CD8α-positive cells per mm^2^ ((**F**), Mann–Whitney test, * *p* < 0.05).

**Figure 7 cancers-16-01901-f007:**
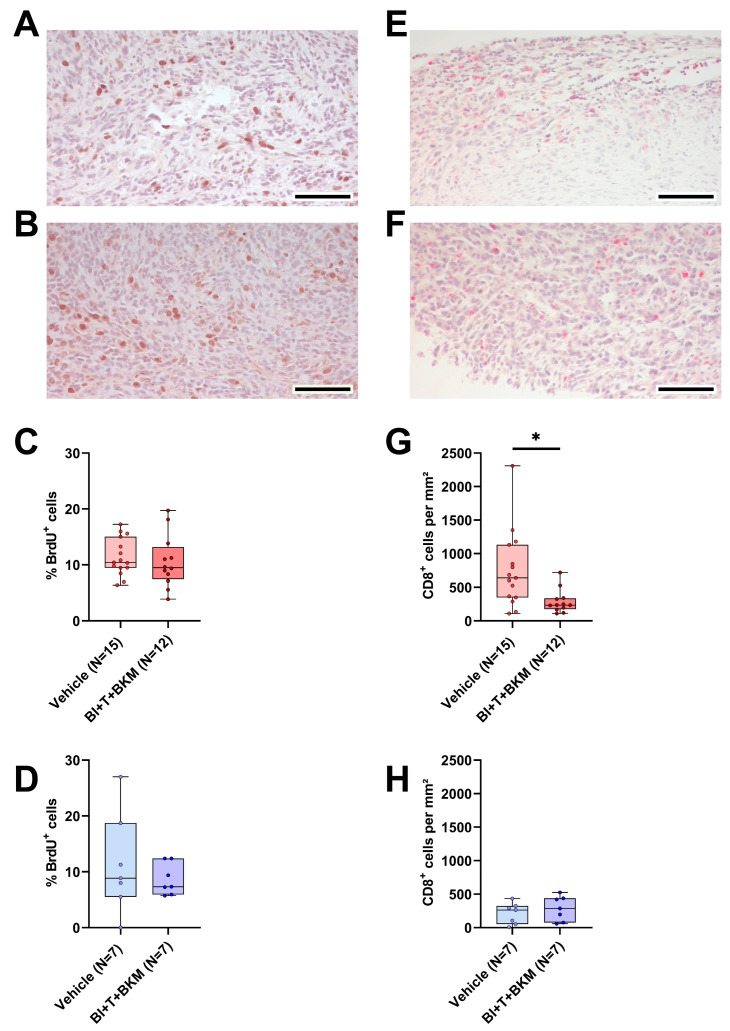
Effect of combinatorial therapy on tumor cell proliferation and amount of intratumoral CD8^+^ cells. (**A**,**B**): Representative histological sections of anti-BrdU stained (brown nuclei) tumors of surviving female (**A**) and male (**B**) mice receiving a combination of drugs consisting of BI-3406 (BI), trametinib (T) and BKM120 (BKM) (scale bar = 100 µm) and the quantitative analysis of the percentage of BrdU^+^ cells in surviving female ((**C**), unpaired *t*-test, ns) and male ((**D**), unpaired *t*-test, ns) mice treated with these drugs or vehicle. (**E**,**F**): Representative histological sections of anti-CD8α stained (red) tumors of surviving female (**E**) and male (**F**) mice receiving these drugs (scale bar = 100 µm) and the quantitative analysis of CD8α-positive cells per mm^2^ in female ((**G**), Mann–Whitney test, * *p* < 0.05) and male ((**H**), unpaired *t*-test, ns) mice treated with drugs or vehicle.

**Table 1 cancers-16-01901-t001:** Measured concentrations (LC-MS/MS) of drugs in the tumor (median with 5–95% confidence interval).

Male (N = 4)	BI-3406 (µM)	Trametinib (µM)	BKM120 (µM)
Median (5–95% CI)	2.4 (0.12–9.3)	0.21 (0.02–0.65)	2.1 (0.68–3.6)
**Female (N = 6)**	**BI-3406 (µM)**	**Trametinib (µM)**	**BKM120 (µM)**
Median (5–95% CI)	3.8 (0.16–38)	0.16 (0.01–1.4)	2.4 (0.41–24)

**Table 2 cancers-16-01901-t002:** Used concentrations of drugs in vitro.

Concentrations used in combination	BI-3406 (µM)	Trametinib (µM)	BKM120 (µM)
	10	0.064	1

## Data Availability

Data will be uploaded to figshare.com; https://doi.org/10.6084/m9.figshare.25144640.

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
