# Peer review of "Sex Matters–Insights from Testing Drug Efficacy in an Animal Model of Pancreatic Cancer"

_cancers, 2024, doi:10.3390/cancers16101901_

Round 1

Reviewer 1 Report (Previous Reviewer 2)

Comments and Suggestions for Authors

The manuscript has been improved according to my previous comments.

Reviewer 2 Report (Previous Reviewer 1)

Comments and Suggestions for Authors

The authors have address all my concerns and I have no further question.

This manuscript is a resubmission of an earlier submission. The following is a list of the peer review reports and author responses from that submission.

Round 1

Reviewer 1 Report

Comments and Suggestions for Authors

The manuscript investigates the impact of sex on PDAC in C57BL6/J animal model and assesses the efficacy and potential side effects of a combinatorial therapy (BI-3406, trametinib, and BKM120). They noticed that male mice exhibited poorer survival, larger tumor sizes, increased metastasis, and higher distress levels compared to females. Despite larger tumors in males, tumor cell proliferation rates were similar between sexes. Then, the study explored potential immunological responses and discovered a distinct immune reaction with more CD8+ cells in female tumors.Next, they found the therapy efficiently inhibited cancer cell proliferation synergistically in vitro. In vivo, the therapy reduced tumor weight more effectively in males, along with a modest decrease in invasiveness and increased survival probability. Additionally, they evaluated the adverse side effects and found that the therapy demonstrated good tolerability, with no significant impact on liver, kidney, or tissue damage. However, they observed an increase in c-peptide concentrations. While drug concentrations in tumors showed no significant difference between sexes, the therapy exhibited sex-specific effects, potentially related to immunomodulatory actions. Overall, the combinatorial therapy shows promise in reducing tumor burden, and its tolerability supports further investigation. However, they emphasize the need to study sex-specific effects in females. The study highlights the importance of considering sex-specific factors in cancer research and therapy development. To enhance the findings of the manuscript, please consider including the following aspects:

1) Discuss potential translational implications, limitations, challenges, and future directions from murine animal model to clinical settings in the context of human pancreatic more explicitly.

2) Include additional experiments or a more thorough review from previous studies to explore the molecular and cellular mechanisms underlying the observed sex-specific effects in response to the therapy in a more in-depth way.

3) Conduct a more comprehensive pharmacokinetic analysis by including measurements of drug concentrations over time, in both male and female mice to elucidate potential differences in drug metabolism, distribution, and elimination between the sexes

4) Besides CD8+ cells, explore the expression levels of other immune checkpoints to analyze the composition and activation status of immune cell populations within the tumor microenvironment.

5) Explore the potential influence of sex hormones by performing experiments involving hormonal manipulations in vitro.

Reviewer 2 Report

Comments and Suggestions for Authors

A manuscript by Schulz et al "Sex Matters – Insights from Testing Drug Efficacy in an Animal Model of Pancreatic Cancer" highlights the importance of sex inclusive research while simultaneously providing a basis for further studies of the therapy tested. The Authors evaluated a model of syngeneic, orthotopic, metastasized pancreatic ductal adenocarcinoma with regard to the impact of sex on pathological features of this disease as well as on the efficacy and possible adverse side effects of a novel, small molecule-based therapy inhibiting KRAS:SOS1, MEK1/2 and PI3K signaling in male and female C57BL6/J mice.

Specific comments:

1. English should revised throughout the manuscript.

2. The reason for using drugs in indicated concentrations should be more clearly explained in the manuscript, in the results section.

3. Fig. 5 - p values should be shown for each comparison even if the differences are not statistically significant.

4. The Authors should more broadly discuss potential limitations of their study with regard substantial differences in N between sex groups e.g., 7 vs 15 (Fig. 6).

Comments on the Quality of English Language

revision required